# Scaling of Yu-Shiba-Rusinov energies in the weak-coupling Kondo regime

Nino Hatter[1], Benjamin W. Heinrich ᴵᴰ [1], Daniela Rolf[1] & Katharina J. Franke[1]

The competition of the free-spin state of a paramagnetic impurity on a superconductor with its screened counterpart is characterized by the energy scale of Kondo screening compared to the superconducting pairing energy $\Delta$. When the experimental temperature suppresses Kondo screening, but preserves superconductivity, i.e., when $\Delta/k_B > T > T_K$ ($k_B$ is Boltzmann's constant and $T_K$ the Kondo temperature), this description fails. Here, we explore this temperature range in a set of manganese phthalocyanine molecules decorated with ammonia on Pb(111). We show that these molecules suffice the required energy conditions by exhibiting weak-coupling Kondo resonances. We correlate the Yu-Shiba-Rusinov bound states energy inside the superconducting gap with the intensity of the Kondo resonance. The observed correlation follows the expectations for a classical spin on a superconductor. This finding is important in view of many theoretical predictions using a classical spin model, in particular for the description of Majorana bound states in magnetic nanostructures on superconducting substrates.

[1] Fachbereich Physik, Freie Universität Berlin, Arnimallee 14, 14195 Berlin, Germany. Correspondence and requests for materials should be addressed to B.W.H. (email: bheinrich@physik.fu-berlin.de)

A single magnetic atom/molecule adsorbed on a super-conductor presents a local perturbation for the quasi-particles with a Coulomb and an exchange scattering potential. The exchange coupling $J$ leads to the formation of localized bound states inside the superconducting energy gap $\Delta$. These so-called Yu-Shiba-Rusinov (YSR)[1–3] states can be detected by tunneling spectroscopy as a pair of resonances symmetrically around the Fermi level ($E_F$)[4–6]. The simplest description of the scattering assumes a classical spin $S$, where the bound-state energy $\varepsilon$ then depends on the coupling strength $J\rho_0$ ($\rho_0$ is the density of states at $E_F$ in the normal state). However, in many cases, a classical description is insufficient. The quantum mechanical nature of the spin manifests in a Kondo resonance outside the superconducting energy gap and in the normal state. Because both YSR and Kondo states are a result of the same exchange coupling strength $J\rho_0$, their energies are connected with each other by a universal relation[7–9]. The formation of the Kondo singlet with its energy scale given by $k_B T_K$ (with $k_B$ being the Boltzmann constant and $T_K$ the Kondo temperature) thereby competes with the singlet state of the superconductor. A quantum phase transition from an unscreened, free-spin ground state to a Kondo-screened state occurs at $k_B T_K \sim \Delta$. Recent experiments corroborated the theoretically predicted universal relation between the YSR bound-state energy $\varepsilon$ and $T_K$[10–14].

An intriguing situation arises if the Kondo energy scale is ill-defined. This is the case when the thermal energy is larger than the energy scale of the Kondo screening. Then, the exchange coupling $J$ gives rise to scattering processes, which induce a zero-bias resonance in transport measurements/tunneling experiments. The width of the resonance is only given by the experimental temperature, i.e., it is independent of $J$. The scattering processes can be well captured within perturbation theory. This description is commonly referred to as weak-coupling Kondo[15, 16]. Contrary to the temperature-dependent Kondo description, the energy of the YSR bound state is temperature independent. The relation between YSR bound-state energy $\varepsilon$ and observables of the weak-coupling Kondo resonance has not been established to date.

Here, we experimentally deduce a new expression of the universal relation between exchange scattering processes in the weak-coupling Kondo regime with the bound-state energy $\varepsilon$ of the YSR states. We show how the height $a$ of the zero-bias resonance correlates with the binding energy $\varepsilon$. The unraveling of this correlation demands for an ensemble with a variety of coupling strengths $J$, all being in the weak-coupling Kondo regime. An ideal system is a Moiré pattern of adsorbates on a super-conductor, where each adsorbate bears a slightly different exchange potential for the substrates' quasiparticles. Manganese phthalocyanine (MnPc) molecules on a Pb(111) surface exhibit such a Moiré pattern with a strong variation in $J$, but the scattering is not in the weak-coupling Kondo regime[11, 17, 18]. We use MnPc/Pb(111) as a template and attach an additional $NH_3$ ligand to the Mn ion, borrowing a successful strategy from surface chemistry. The surface trans-effect reduces the coupling to the substrate[19, 20] while the variety of adsorption sites is maintained. We show that all these molecules exhibit a weak-coupling Kondo resonance and YSR states, where the height of the Kondo resonance and the YSR energy are connected via the exchange coupling strength $J\rho_0$.

## Results

**Topographic appearance and characterization of MnPc–NH₃.**
Figure 1a shows a typical scanning tunneling microscopy (STM) topography of an MnPc island after $NH_3$ adsorption. While the square-like structure of the MnPc monolayer island is preserved, the appearance of the molecules is altered compared to the characteristic clover-shape pattern of the pristine MnPc molecule (see inset of Fig. 1a for a zoom on seven MnPc–NH₃ and two MnPc molecules). MnPc–NH₃ molecules appear nearly circular symmetric and $3.1 \pm 0.2$ Å high, i.e., $\approx 1.7 \pm 0.2$ Å higher than MnPc. We interpret this in terms of $NH_3$ binding to the central Mn ion and protruding toward the vacuum. By means of voltage pulses, we can desorb the $NH_3$ from the molecule in a controlled manner and restore pristine MnPc (see Supplementary Fig. 1 and Supplementary Note 1).

**YSR bound states as indicators of magnetic coupling strength.**
To study the exchange coupling strength between the molecules and the superconductor, we perform $dI/dV$ spectroscopy with a superconducting tip (see Methods section for details). Figure 1b (black curve) shows a characteristic example of a $dI/dV$ spectrum on MnPc–NH₃. The spectrum is dominated by the

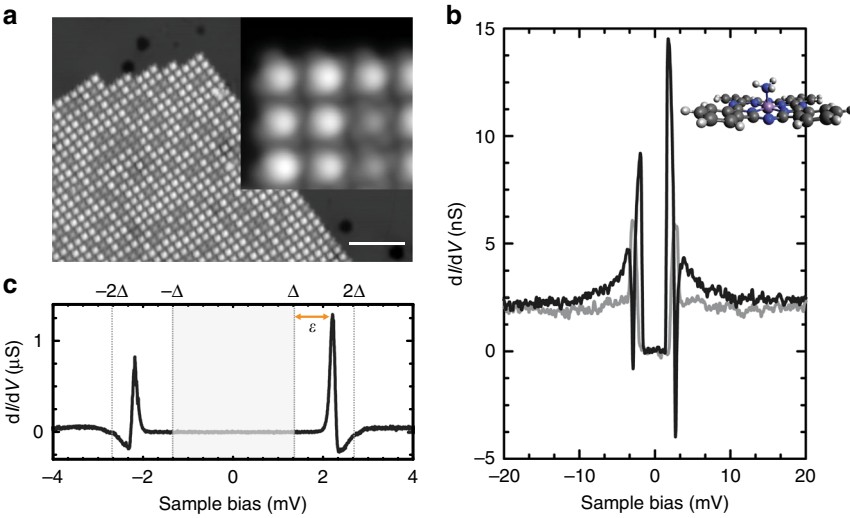

**Fig. 1** MnPc–NH₃ on Pb(111). **a** Topography of a mixed island of MnPc–NH₃ and MnPc molecules (setpoint: $U = 50$ mV, $I = 100$ pA; scale bar is 10 nm). The inset shows a zoom on seven MnPc–NH₃ and two MnPc molecules (50 mV, 200 pA). **b** Characteristic $dI/dV$ spectrum of an MnPc–NH₃ (black) and on the bare Pb(111) (gray) acquired with a superconducting Pb tip (50 mV, 200 pA, $U_{mod} = 500$ μV$_{rms}$). **c** Zoom on the superconducting gap presenting a pair of YSR resonances (5 mV, 200 pA, 20 μV$_{rms}$). The bound-state energy is indicated by $\varepsilon$ and the tip gap by the gray area

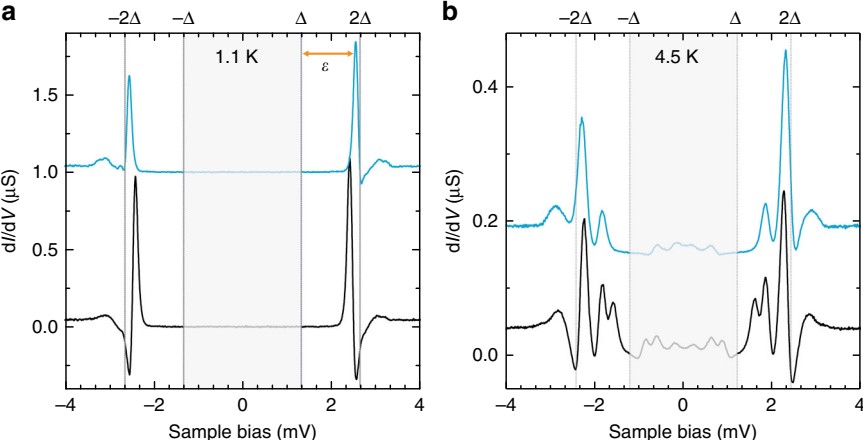

**Fig. 2** YSR excitations at different temperatures. Two typical spectra of MnPc–NH$_3$ at 1.1 K (**a**) and 4.5 K (**b**), respectively, acquired with a superconducting Pb tip (5 mV, 200 pA, 20 μV$_{rms}$). Gray-shaded areas indicate sample biases $|eV| \leq \Delta_{tip}$. At 4.5 K (**b**), additional resonances are observed because of tunneling out off/into thermally excited YSR states

superconducting gap structure in the range of $\approx \pm 3$ meV around the Fermi level $E_F$. At energies outside the superconducting gap, an increasing intensity is observed when approaching $E_F$ from both bias sides. In the later part of this manuscript, we will show that this can be associated with a zero-bias resonance due to Kondo scattering.

Within the superconducting gap, a single pair of YSR resonances is observed (high-resolution zoom in Fig. 1c). The electron-like component, i.e., the resonance at positive energy, is more intense than the hole-like resonance. This holds true for all MnPc–NH$_3$ molecules studied (see Supplementary Fig. 2d). We have studied a set of 44 molecules before and after controlled, tip-induced desorption of the NH$_3$ ligand. The comparison allows us to assign the bound-state energy $\varepsilon$ to the intense electron-like excitation, i.e., $\varepsilon > 0$. Hence, all MnPc–NH$_3$ molecules are in the unscreened, free-spin ground state (see Supplementary Fig. 2 and Supplementary Note 2 for details).

Variations in $\varepsilon$ for different molecules are observed, but $\varepsilon$ is always more positive than in the case of pristine MnPc. The small variations in $\varepsilon$ of MnPc–NH$_3$ correlate with the stronger variations in $\varepsilon$ of the pristine MnPc molecules, i.e., after NH$_3$ desorption. Hence, the coupling strength to the substrate electrons $J$ again varies within the Moiré pattern, but is reduced compared to MnPc as expected for the surface trans-effect[19, 20]. The addition of the axial NH$_3$ ligand on the central ion pulls the Mn away from the surface and weakens the Mn–surface coupling.

Unambiguous evidence for the free-spin ground state is found by measurements at a slightly elevated temperature. While at 1.1 K, we observe only a single pair of YSR resonances for all MnPc–NH$_3$ molecules, we observe two or three pairs of resonances together with their thermal replica for all complexes at 4.5 K (see two characteristic examples in Fig. 2). The separation of these resonances amounts to 200–400 μeV. In most cases, this splitting is larger than the splitting of the resonance in the case of pristine MnPc. The absence of these resonances at lower temperature shows that they are linked to a thermally activated occupation of low-lying excited states. This behavior signifies an anisotropy-split ground state[18], which can be explained by a free spin $S = 1$ with axial and transversal anisotropy[21]. We emphasize that a splitting of the excited YSR state would result in a set of three YSR pairs independent of the experimental temperature[18]. Note that, unlike in the case of iron phthalocyanine on Au(111), where NH$_3$ adsorption induces a spin change from an intermediate spin state of $S = 1$ to a low spin of $S = 0$[22], we do not find an indication for a change of the spin state here.

**Kondo effect in the weak-coupling regime**. Next, we analyze the resonance outside the superconducting gap, which was already identified in Fig. 1b. For this, we quench the superconducting state of the substrate by applying a magnetic field of $B = 0.1$ T perpendicular to the sample surface and we employ a normal metal tip. Figure 3a shows typical spectra acquired on MnPc–NH$_3$ at various temperatures when the substrate is in the normal state. A zero-bias resonance is observed, which is reminiscent of the Kondo effect. At low temperatures, the peak is split around $E_F$.

We first focus on this splitting, which cannot be explained in terms of Zeeman energy (~11 μeV for $g \approx 2$), because the field strength is low compared to temperature ($k_B T \approx 100$ μeV at 1.1 K). To explore a possible magnetic origin of the splitting, we record d$I$/d$V$ spectra on one molecule in fields up to 3 T. While the overall shape of the zero-bias resonance does not change (Supplementary Fig. 3 and Supplementary Note 3), a zoom on the dip at $E_F$ unveils an opening of the gap with increasing out-of-plane field (Fig. 3b). Hence, the gap can be associated to inelastic spin excitations. The extracted step energies (Fig. 3e) are fit to a simple Spin Hamiltonian, which assumes the anisotropy axis being parallel to the out-of-plane field: $\hat{\mathcal{H}} = DS_z^2 - g\mu_B B_z S_z$. Here, $D$, $S_z$, $B_z$, $\mu_B$, and $g$ are the axial anisotropy parameter, the projection of the spin and the magnetic field in $z$ direction, the Bohr magneton, and the Landé $g$-factor, respectively. For a spin of $S = 1$, $D$ amounts to $-0.33 \pm 0.01$ meV and $g$ is $2.0 \pm 0.1$.

It is noteworthy that, on other MnPc–NH$_3$ molecules, we observed a linear dependence of the step energies on field strength only above $\approx 1$ T and a slower increase at lower fields. In these cases, the $B$-field dependence is well reproduced when accounting for an additional in-plane anisotropy term $E(S_x^2 - S_y^2)$ in the Hamiltonian (see Supplementary Fig. 6 and Supplementary Note 6). Occasionally, we also observe two pairs of excitation steps as is expected in the case of non-zero $E$ (Supplementary Fig. 5 and Supplementary Note 5). Yet, most of the times, this is blurred by the limited energy resolution ($\approx 300$ μeV at 1.1 K).

The aforementioned observation of inelastic spin excitations fit to an $S = 1$ spin system with dominant axial anisotropy and are in line with the measurements in the superconducting state. The YSR resonances are split into two (three) resonances separated by up to 400 μeV, when measured at 4.5 K, which indicates $E \approx 0$ ($E \neq 0$).

Interestingly, the zero-field splitting is observed on top of a zero-bias resonance with a half-width at half-maximum (HWHM) of 3.7 meV. In order to split a (strong-coupling)

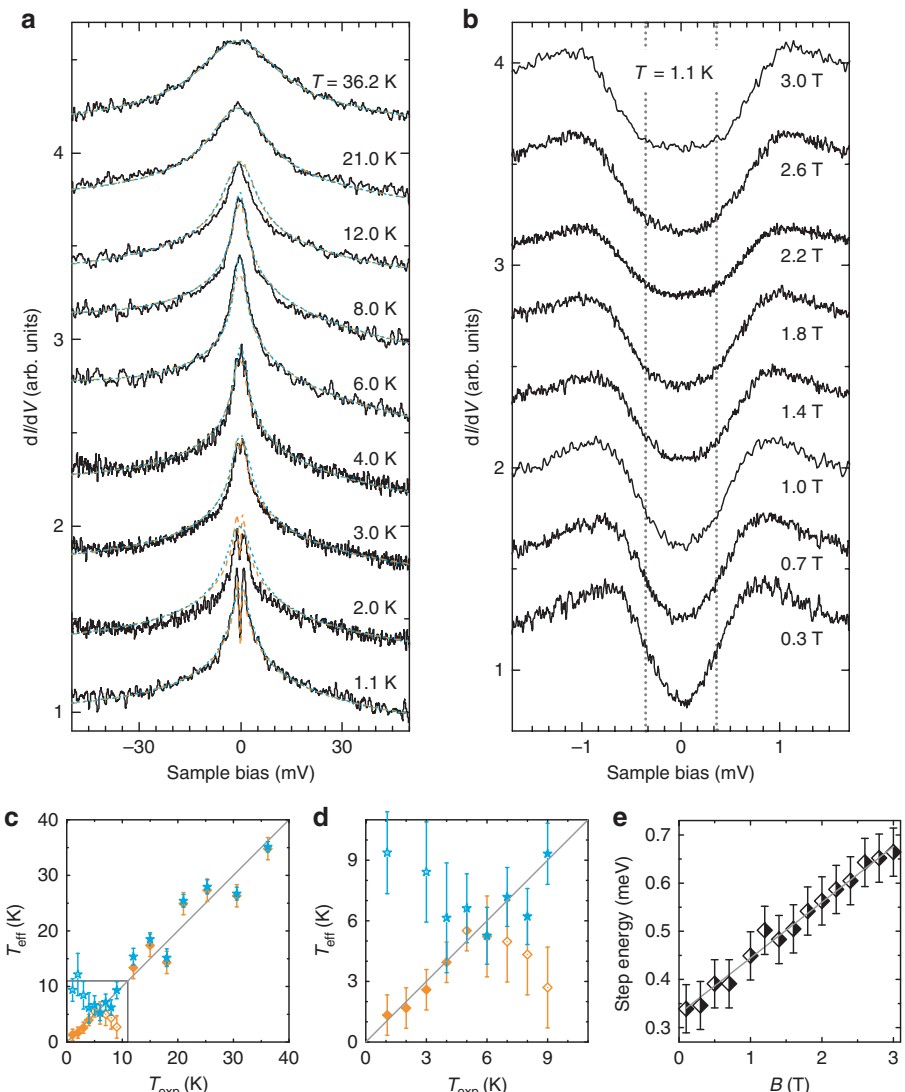

**Fig. 3** Temperature and *B*-field dependence of the Kondo resonance. **a** Temperature evolution of the zero-energy resonance at $B = 0.1$ T, which quenches the superconductivity in the sample (50 mV, 200 pA, 500 $\mu V_{rms}$). The spectra are offset for clarity. A split zero-energy resonance is observed. Orange (blue) lines are fits according to the third-order scattering model as described in refs. [15, 16] including (excluding) uniaxial magnetic anisotropy as determined in (**e**). **b** Evolution of the step-like feature around $E_F$ in an external *B* field parallel to the surface normal (5 mV, 200 pA, 50 $\mu V_{rms}$). **c, d** Effective temperature $T_{eff}$ as extracted from fits like in (**a**) versus the experimental temperature $T_{exp}$. Orange squares (blue stars) are from fits accounting for (disregarding) the axial anisotropy as determined in (**e**). **e** Step energies versus *B*. Energies are extracted from the d*I*/d*V* spectra in (**b**) by a least-squares fit with symmetric step functions. The gray line indicates the fit to a Spin Hamiltonian for $S = 1$ with uniaxial anisotropy: $D = -0.33 \pm 0.01$ meV; $g = 2.03 \pm 0.07$. All spectra were acquired with a normal metal Au tip. Error bars in **c**–**e** account for the standard error on the least-squares fit parameter

Kondo resonance of this width by means of an external magnetic field, a critical field of $B_c \approx 65$ T would be needed. Yet, the axial anisotropy of 0.3 meV—this is equivalent in energy to a field of 3 T—is sufficient to induce a sizable splitting. The absence of a critical field contradicts an explanation of the zero-bias resonance as an expression of a strong-coupling Kondo effect. However, the zero-bias resonance agrees with the system being in the weak-coupling Kondo regime.

To corroborate this interpretation, we performed temperature-dependent measurements (Fig. 3a). With increasing temperature, the height of the resonance decreases and its width increases. The symmetric steps close to the Fermi level broaden and vanish at $T \approx 4$ K. Following our arguments derived from the *B*-field dependence, the zero-bias resonance in the d*I*/d*V* spectra shall be described by scattering at the impurity spin during tunneling. The formalism of the weak-coupling Kondo effect has been

described by Anderson and Appelbaum[23–25] within the second-order perturbation theory, which accounts for the third-order scattering processes. To test this scenario, we fit the spectra using the scattering approach as described in ref. [16], accounting for the axial anisotropy as determined above (orange lines in Fig. 3a). The fit accounts for a broadening of the logarithmic zero-bias divergence by an effective temperature parameter $T_{eff}$. In Fig. 3c, we draw $T_{eff}$ against the experimental temperature $T_{exp}$. In a broad range of $T_{exp}$, the values fall onto the identity line (orange squares in Fig. 3c), which is a good evidence for the weak-coupling Kondo regime.

We note that in the temperature range of 6–9 K, $T_{eff}$ drops below $T_{exp}$. Interestingly, the deviation starts when the thermal energy $k_B T$ surpasses the anisotropy energy of 330 $\mu$eV. Apparently, the thermal scattering diminishes the effect of the magnetocrystalline anisotropy. We fit the d*I*/d*V* spectra again

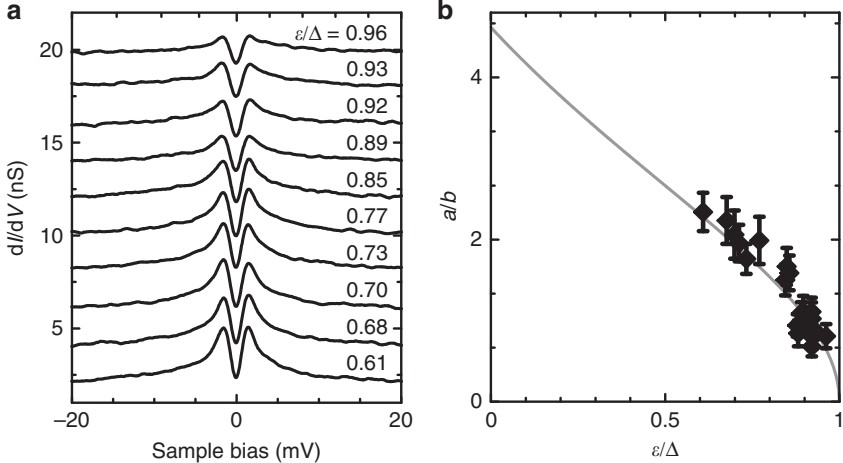

**Fig. 4** Kondo scattering versus YSR energy. **a** Selected $dI/dV$ spectra of different MnPc–NH₃ complexes ordered according to the YSR energy (50 mV, 200 pA, 500 μV$_{rms}$). Spectra are acquired at $B = 2.7$ T in order to quench the superconductivity in the Pb-covered tip and the sample. The spectra are offset for clarity. **b** Amplitude over background $a/b$ as a function of $\varepsilon/\Delta$. $a/b$ is extracted from the spectra as in (**a**). $\varepsilon/\Delta$ as extracted from the spectra at $B = 0$ T similar to Fig. 1c. Error bars indicate the uncertainty of the read out. For $\varepsilon$, the error is in the order of the symbol size

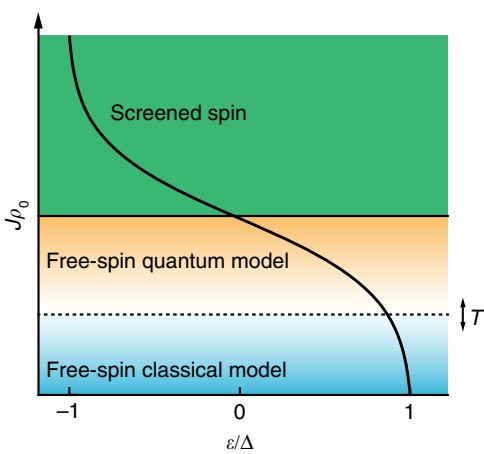

**Fig. 5** Sketch of the YSR phase diagram. The transition from a classical spin model to a quantum description depends on the temperature

with the scattering approach, but this time without any anisotropy term (blue stars). Then, $T_{eff}$ is larger than $T_{exp}$ below 4 K, but falls on the identity line above. Hence, these experiments show how magnetocrystalline anisotropy can affect third-order scattering signatures, but that the effect is diminished by temperature.

We also performed a fit with a Fano-Frota function including temperature broadening (see Supplementary Fig. 4 and Supplementary Note 4). The analysis corroborates that the system is not in the strong-coupling Kondo regime.

**Correlation of weak-coupling Kondo resonance with YSR energy**. Next, we attempt to correlate both expressions of scattering at the magnetic adsorbate, i.e., YSR and weak-coupling Kondo physics, in order to establish a universal picture. Both, the third-order scattering and the YSR bound states depend on the exchange coupling strength $J\rho_0$. Assuming a classical spin, the bound-state energy $\varepsilon$ was predicted to scale as $\varepsilon/\Delta = (1 - \alpha^2)/(1 + \alpha^2)$, with $\alpha = \pi S J \rho_0$ [2,3].

Using the second-order perturbation theory, the amplitude $a$ of the zero energy resonance caused by third-order spin scattering scales with $(J\rho_0)^3$ [15]. The apparent height of the molecule of 3.1 Å

prohibits direct tunneling to the substrate. The symmetric, logarithmic shape of the zero-bias resonance—opposed to an asymmetric Fano line shape caused by the interference of two tunneling paths—indicates that the background conductance $b$ is dominated by second-order spin scattering (elastic and inelastic) and scales with $(J\rho_0)^2$ (see ref. [26] and Supplementary Information of ref. [15]). Then we can approximate:

$$\frac{a}{b} \propto \frac{1}{\pi S} \sqrt{\frac{1 - \varepsilon/\Delta}{1 + \varepsilon/\Delta}}. \tag{1}$$

The variations in $J\rho_0$ caused by the different adsorption sites of the molecules within the Moiré allow us to test this relation. We first determine $\varepsilon$ of different complexes using a superconducting tip. We then measure $dI/dV$ spectra of the zero-bias resonance on the same molecules at $B = 2.7$ T, which is necessary in order to quench the superconductivity also in the tip, which has an increased critical field because of its finite size. We then can extract the amplitude $a$. Figure 4a shows a selection of such spectra ordered according to their YSR energy $\varepsilon$. With increasing $\varepsilon$, the amplitude $a$ decreases. In Fig. 4b, we plot the amplitude-over-background ratio $a/b$ as a function of YSR energy $\varepsilon$ over $\Delta$. A fit to Eq. (1) with a linear scaling as the only free parameter is shown in gray and describes the data evolution well. Hence, we conclude that the description of a YSR impurity as a classical spin is a sufficient model in the case of the Kondo effect in the weak-coupling regime.

## Discussion

The exchange coupling strength $J$ controls the magnetic properties of an impurity on a superconductor. It determines whether the adsorbate maintains a spin and, therefore, a magnetic moment, or whether the spin is screened. Usually, the exchange coupling strength cannot be measured directly, albeit its impact is seen in Kondo resonances and YSR states. Because $J$ is responsible for both, the energy scale of the Kondo effect and the energy of the YSR states, there exists a universal relation between these phenomena [7–9]. Typically, this relation is discussed at $T = 0$ (or at least at $T \ll T_K$), neglecting the energy scale of the experimental temperature. Though, temperature plays an important role. In particular, it can drive the crossover from a coherent, Kondo-

screened spin state at $T \ll T_K$ to a state at $T \gg T_K$, where a coherent many-body ground state is absent and spin-scattering processes can be described to some leading order in perturbation theory.

In contrast to Kondo physics, the energy of YSR states is not influenced by temperature (since $\Delta \approx$ const. for $T_{exp} \lesssim T_c/2$). However, it depends on the energy scale of the Kondo effect. Therefore, an understanding of the regime $T \gg T_K$ is necessary in order to fully capture all relevant energy scales. A priori, it was not obvious that the YSR resonances can be treated classically in this regime. Our experiments reveal a relation to the exchange coupling $J\rho_0$, as expected intuitively, and a good agreement with the treatment in the classical limit. We note that a free spin is not a sufficient condition for a classical description. The free-spin regime also exists in the limit of $T \ll T_K$, where the quantum mechanical description is required. Hence, we add another regime to the well-known phase diagram of magnetic impurities on superconductors (sketch in Fig. 5). Contrary to the transition between the screened-spin and free-spin state, this transition depends on temperature. Our results thus conclude with a picture of all the necessary energy scales.

This is of interest not only for the single-impurity problem as discussed here, but also for the exotic physics of magnetic, nanoscale structures on superconductors. Their theoretical description often relies on classical spin models, because of the quantum impurity model being impossible to treat with analytical methods[27]. In particular, the suggestion of Majorana bound states in magnetic impurity chains and arrays on s-wave superconductors has been put forward in the classical spin description[28–38].

## Methods

**Sample preparation**. The Pb(111) single crystal surface was thoroughly cleaned by cycles of Ne$^+$-ion sputtering and subsequent annealing to 430 K. MnPc was deposited from a Knudsen cell at $\approx$710 K onto the clean Pb(111) surface held at room temperature. Subsequently, 3 to 6 Langmuir of Ammonia (NH$_3$) were dosed onto the as-prepared sample cooled to $\approx$15 K. In order to desorb excess ammonia, the sample was annealed to $\approx$100 K for several hours. This procedure ensured that $\approx$90% of the MnPc molecules are coordinated by a single NH$_3$ molecule, while the Pb(111) surface was free of NH$_3$.

**Preparation of the superconducting tip**. Pb-covered, superconducting tips for high–resolution spectra were obtained by macroscopic indentations of a W tip into the clean Pb sample following ref. [11]. We used only tips, which showed bulk-like superconductivity, i.e., where the gap parameter of the tip $\Delta_{tip}$ was equal to the sample's gap parameter $\Delta_{sample}$ within our resolution. Hence, we can write $\Delta_{tip} \cong \Delta_{sample} \cong \Delta$. In d$I$/d$V$ spectra, the superconducting tip then shifts all energies by $\pm\Delta$ and the gap size amounts to $4\Delta$ with the quasiparticle excitation peaks on the pristine surface at $\pm 2\Delta$. See the Supplementary Information of ref. [37] for more details on the determination of $\Delta_{tip}$.

**Measurement details**. All measurements were performed in a Specs JT-STM at a temperature of 1.1 K, if not stated differently. Measurements of the sample in the normal metal state were performed at 0.1 T with an Au-covered W tip (Fig. 3), or with a Pb-covered tip at 2.7 T (Fig. 4). Spectra of the differential conductance d$I$/d$V$ were acquired with standard lock-in technique at a frequency of 912 Hz and a root-mean-square (rms) bias modulation $V_{mod}$ as indicated in the figure captions.

**Data availability**. The main data that support the findings of this study are available within the paper (including the Supplementary Information). Additional source data for Figs. 3e and 4b and Supplementary Fig. 6 are available from B.W.H. (bheinrich@physik.fu-berlin.de) on reasonable request.

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

## Acknowledgements

We thank F. von Oppen, J.I. Pascual, and M. Ternes for enlightening discussions. We acknowledge funding by the Deutsche Forschungsgemeinschaft through Grant Nos. FR2726/4 and HE7368/2 as well as by the European Research Council through Consolidator Grant NanoSpin.

## Author contributions

B.W.H. and K.J.F. designed the research. N.H. did the measurements with the help of B.W.H. and D.R. All authors discussed the results. N.H., B.W.H., and K.J.F. wrote the paper with inputs from D.R.

## Additional information

**Competing interests:** The authors declare no competing financial interests.

