## [Peer Review File · Nature Communications]

Reviewer #1 (Remarks to the Author):

The interaction between local spin with metallic host and the superconductivity are of long-term interests for the many body physics. Authors studied the YSR state and Kondo resonance of manganese phthalocyanine molecules decorated with ammonia (MnPc-NH₃) on Pb(111) at weak coupling Kondo regime. They discovered the correlation between the energy of the YSR states in the superconducting energy gap and the intensity of the Kondo resonance. They found that, in this weak coupling Kondo regime, the local spin could be treated as a classical one and the energy of YSR states follows the theoretical expectation. Authors show very nice experiments with high quality data. The theoretical analysis excellently explains the experimental results, which is convincing to me. The results of this work are of significant progress in this field, and would undoubtedly generate a broad interests in the fundamental physics. I recommend the publication of this excellent work on Nature Communications. For some following issues, I hope authors could address them clearly.

(1) Authors assume the background conductance b to be mainly (> 95%) caused by second order spin scattering (elastic and inelastic). How could authors give such estimation of 95%?

(2) The reference (Nature Communications 4, 2110 (2013)) points out the steps due to spin-flip excitations is proportional to $(J\rho_0)^2$. In my understanding, the background conductance in current experiment may also be contributed by the molecular orbitals, which may not relate to spins. So to me, the background conductance may be not proportional to $(J\rho_0)^2$. The assumption of background b should be carefully addressed.

Reviewer #2 (Remarks to the Author):

The manuscript by Hatter and coworkers uses low temperature scanning tunnelling spectroscopy to study Yu-Shiba-Rusinov (YSR) states and the Kondo resonance of Phthalocyanine molecules. The authors have acquired spectra in the normal and the superconducting state on top of the molecules. They use these results to show that the Kondo effect on the molecules is in the weak coupling limit and to relate the Kondo coupling $J\rho_0$ to the parameters of the YSR resonances, and show that a classical treatment of the spin is sufficient to capture the physics in this regime. The paper is well written, sound and interesting. I would therefore recommend publication in Nature Communications with only a few minor corrections as listed below.

* The authors should clearly define how Δ and 2Δ as indicated in their figures are determined

* It would be good if the authors added a sentence to explain what the "surface trans effect" is

* Fig. 2: please properly explain the inset. The panel in the inset is very small, so it might be better to show this on a separate panel of the figure. The caption doesn't even mention the inset.

* For fig. 3b, it would be good to specify the temperature since panels a, c and d show temperature dependences. (while the methods section says the temperature of the measurement is 1.1K unless otherwise stated, this is not necessarily clear for fig. 3b)

* Y-axis labels in fig S4b seem to be out of place

We thank the referees for their positive feedback and for pointing out important points, which were not sufficiently clear in the manuscript.

Reply to Referee 1:

- 1) *Authors assume the background conductance b to be mainly (> 95%) caused by second order spin scattering (elastic and inelastic). How could authors give such estimation of 95%?*

Our estimation of at least 95% of second order spin scattering in the background conductance is based on a lineshape analysis of the weak-coupling Kondo resonance. We observe a symmetric logarithmic peak. Ref. 15 (Nature Communications 4, 2110 (2013) and Supplementary Note 3 therein) points out that the interference between different tunneling channels may lead to a Fano lineshape of the weak-coupling Kondo resonance, depending on the ratio of the tunneling paths (similar to the better studied case of the resonance lineshape in the strong-coupling Kondo regime). A dip would indicate dominant tunneling into substrate (or molecular) state, while the logarithmic peak observed in the experiment evidences the dominant tunneling via the impurity d state.

Based on this argument, the background conductance is dominated by spin scattering processes (elastic and inelastic). While also fits to the spectra pointed to a realistic estimation of 95%, we have removed this value in the manuscript, because the fit was overdetermined in parameters. Therefore, we refrain from giving a numerical value, but clarify our arguments for the dominant contribution of second order spin scattering in the background. We have added an explanation and two references in the manuscript:

“The symmetric, logarithmic shape of the zero bias resonance - opposed to an asymmetric Fano line shape caused by interference of two tunneling paths - indicates that the background conductance b is dominated by second order spin scattering (elastic and inelastic) and scales with $(J\rho_0)^2$ (see Ref. [26] and Supplementary Information of Ref. [15]).”

- 2) *The reference (Nature Communications 4, 2110 (2013)) points out the steps due to spin-flip excitations is proportional to $(Jr_0)^2$. In my understanding, the background conductance in current experiment may also be contributed by the molecular orbitals, which may not relate to spins. So to me, the background conductance may be not proportional to $(Jr_0)^2$. The assumption of background b should be carefully addressed.*

This remark directly relates to the previous point. Based on our argument above, we can assign the background conductance mainly to spin scattering. Yet, we have tested scenarios with different, constant proportions of other contributions to the background conductance (like direct tunneling through molecular states or into the substrate). The result is shown in the attached figure (figure_for_referee.pdf), where we accounted for 10%, 20% and 50% of the maximal measured background (b_{\max}) as additional background conductance. $x=0$ is the case considered in the manuscript. The fit to eq. (1) of the manuscript is getting worse, the more constant background is considered, but can still describe the overall trend. Hence, our conclusions hold also in the unlikely case of substantial co-tunneling.

Reply to Referee 2:

- 1) *The authors should clearly define how Delta and 2Delta as indicated in their figures are determined.*

Throughout the experiments, we used tips, which showed bulk-like superconductivity, i.e., where the gap parameter of the tip Δ_{tip} is similar to the sample's gap parameter Δ_{sample} (for sake of simplicity, we neglect the two-band nature of superconductivity in the single crystal bulk sample here). From dI/dV spectra with superconducting tips, we cannot determine the parameters Δ_{tip} and Δ_{sample} independently. An independent determination of both parameters is only possible using spectra with a pronounced low-energy YSR state, which gives rise to well-resolved thermal resonances (as is the case of some of the MnPc molecules shown in the supplementary Information in Fig. S2). The YSR resonance and its thermal counterpart occur symmetric to Δ_{tip} at $(\Delta_{\text{tip}}+\epsilon)$ and $(\Delta_{\text{tip}}-\epsilon)$. This allows us to determine Δ_{tip} unambiguously.

We then use the same tip to acquire spectra of the pristine surface. These show clear BCS resonances at $\Delta_{\text{tip}}+\Delta_{\text{sample}}$. Because Δ_{sample} is a bulk property of the substrate, the energy can then serve to determine Δ_{tip} for different tips.

As $\Delta_{\text{tip}} \approx \Delta_{\text{sample}} \approx \Delta$, we simplify the nomenclature in the paper and use only Δ and 2Δ instead of Δ_{tip} and $\Delta_{\text{tip}}+\Delta_{\text{sample}}$, respectively. We think that this improves the readability of the paper. We added an explanation to the Methods section of the manuscript:

“We used only tips, which showed bulk-like superconductivity, i.e., where the gap parameter of the tip Δ_{tip} was equal to the sample's gap parameter Δ_{sample} . Hence, we can write $\Delta_{\text{tip}} \cong \Delta_{\text{sample}} \cong \Delta$. In dI/dV spectra, the superconducting tip then shifts all energies by $\pm\Delta$ and the gap size amounts to 4Δ with the quasiparticle excitation peaks on the pristine surface at $\pm 2\Delta$. See the Supplementary Information of Ref. 37 for more details on the determination of Δ_{tip} .”

- 2) *It would be good if the authors added a sentence to explain what the “surface trans effect” is.*

We added an explanation in the manuscript when introducing the surface trans-effect:

“The addition of the axial NH_3 ligand on the central ion pulls the Mn away from the surface and weakens the Mn--surface coupling.”

- 3) *Fig. 2: please properly explain the inset. The panel in the inset is very small, so it might be better to show this on a separate panel of the figure. The caption doesn't even mention the inset.*

The inset shows a collection of 44 spectra acquired on different MnPc-NH₃ molecules. The same panel is also shown in Fig. S2d of the Supplementary Information. Thanks to the referee's comment, we realized that the inset is not useful for the reader. As we judge the information not essential for the understanding of the paper, we removed the inset of Fig. 2a. Yet, we keep it in Fig. S2d, where we explain it in the caption.

- 4) *For fig. 3b, it would be good to specify the temperature since panels a, c and d show temperature dependences. (while the methods section says the temperature of the measurement is 1.1K unless otherwise stated, this is not necessarily clear for fig. 3b).*

We agree that the specific statement of the temperature avoids ambiguity. We changed the figure accordingly.

- 5) *Y-axis labels in fig S4b seem to be out of place.*

We corrected the axis label.

Additional changes to the manuscript.

According to the guideline of Nature Communication, we have removed all footnotes and included them in the text.

We have added a new reference (ref. 27) to the manuscript at the very end of the discussion:

“Their theoretical description often relies on classical spin models because of the quantum impurity model being impossible to treat with analytical methods [27].”

REVIEWERS' COMMENTS:

Reviewer #1 (Remarks to the Author):

Authors have clearly addressed my raised questions. I recommend the publication of the revised manuscript.

Reviewer #2 (Remarks to the Author):

The authors have addressed all my comments as well as the ones of the other referee well. I therefore recommend the manuscript to be accepted for publication in Nature Communications.